# MicroRNA-Mediated Gene Regulatory Mechanisms in Mammalian Female Reproductive Health

**DOI:** 10.3390/ijms22020938

**Published:** 2021-01-19

**Authors:** Samuel Gebremedhn, Asghar Ali, Munir Hossain, Michael Hoelker, Dessie Salilew-Wondim, Russell V. Anthony, Dawit Tesfaye

**Affiliations:** 1Animal Reproduction and Biotechnology Laboratory, Department of Biomedical Sciences, Colorado State University, 3051 Rampart Rd, Fort Collins, CO 80523, USA; Etay@colostate.edu (S.G.); asghar.ali@colostate.edu (A.A.); Russ.Anthony@ColoState.edu (R.V.A.); 2Department of Animal Breeding and Genetics, Bangladesh Agricultural University, Mymensingh 2202, Bangladesh; mmhabg@gmail.com; 3Institute of Animal Sciences, Department of Animal Breeding and Husbandry, University of Bonn, 53115 Bonn, Germany; mhoe@itw.uni-bonn.de (M.H.); dsal@itw.uni-bonn.de (D.S.-W.)

**Keywords:** miRNAs, endometritis, PCOS, ageing, fertility

## Abstract

Mammalian reproductive health affects the entire reproductive cycle starting with the ovarian function through implantation and fetal growth. Various environmental and physiological factors contribute to disturbed reproductive health status leading to infertility problems in mammalian species. In the last couple of decades a significant number of studies have been conducted to investigate the transcriptome of reproductive tissues and organs in relation to the various reproductive health issues including endometritis, polycystic ovarian syndrome (PCOS), intrauterine growth restriction (IUGR), preeclampsia, and various age-associated reproductive disorders. Among others, the post-transcriptional regulation of genes by small noncoding miRNAs contributes to the observed transcriptome dysregulation associated with reproductive pathophysiological conditions. MicroRNAs as a class of non-coding RNAs are also known to be involved in various pathophysiological conditions either in cellular cytoplasm or they can be released to the extracellular fluid via membrane-bounded extracellular vesicles and proteins. The present review summarizes the cellular and extracellular miRNAs and their association with the etiology of major reproductive pathologies including PCOS, endometritis, IUGR and age-associated disorders in various mammalian species.

## 1. Introduction

Micro RNAs (miRNAs) are short noncoding RNA molecules and comprise 1–5% of animal genes [1]. While a typical miRNA could regulate the expression of hundreds of genes, a single target gene can be regulated by multiple miRNA [2]. Altogether they regulate 30% of the active human genes [3]. Changes in the expression of even a single miRNA found to have a profound effect on the outcome of diverse cellular functions. Advancements in library preparation and deep sequencing technologies have enabled the identification of several thousands of miRNAs in various mammalian species, as deposited in the public database, namely miRbase (http://www.mirbase.org). The importance of miRNAs in female fertility has been evidenced by conditional knockout of various miRNA-processing machinery genes using mouse models [4,5]. Accordingly, miRNAs were found to regulate various reproductive processes including germ cells proliferation and differentiation, oocyte growth and maturation, preimplantation embryo development and fetal growth and development. Moreover, miRNA mediated posttranscriptional regulation of epigenetic modifications including DNA methylation, RNA modification, and histone modifications has been reported [6].

Infertility is becoming a global health issue affecting 10–15% of women worldwide. Among the factors contributing to this phenomenon, lifestyle factors and reproductive health issues contribute significantly to infertility problems in humans. Polycystic ovary syndrome (PCOS) is one of the multifactorial reproductive and metabolic health issues that affects up to 5–10% of women in reproductive age and accounts for 75% of anovulatory infertility [7,8,9,10]. PCOS is characterized by follicle growth arrest, reduced granulosa cell proliferation, elevated levels of luteinizing hormone, reduced levels of follicle-stimulating hormone (FSH), and hyperandrogenemia [7]. These endocrine and metabolic imbalances due to inappropriate regulation of the hypothalamus–pituitary–gonadal axis commonly lead to ovulatory disorders in mammals. Different animal models have been used to understand the etiology of PCOS with translation potential to humans [11]. Recent molecular analysis using ovarian cortex, granulosa, and theca cells and follicular fluid has identified complex genetic pathways contributing to the etiology of PCOS.

Bacterial infection in the uterus is known to cause clinical or subclinical endometritis in 15–20% of cows postpartum with a significant impact on fertility. In the post-partum period since cows have multiple confounding conditions, the direct link between reduced fertility and endometritis is practically difficult. However, several in vitro or in vivo controlled experiments have been conducted to investigate the uterine infection-mediated changes in molecular signals in the endometrium and another reproductive tract with a significant impact on ovarian function and embryo implantation [12]. Despite the evidence of involvement of several classes of microorganisms ranging from Gram-negative (*E. coli*) or Gram-positive (*T. pyogenes*) bacteria and several anaerobes in endometritis, there are still controversial results from various model studies involving various pathogens. However, several studies have evidenced the dysregulation of various molecular signals in mammalian endometrium and in circulation, which mediate the negative impact of endometritis on oocyte and embryo growth. Moreover, intrauterine growth restriction (IUGR) as pregnancy-associated disorder remains a major problem in various farm animals and humans resulting in impaired fetal growth. Even though several maternal and fetal factors contribute to the development of IUGR, impaired placental function and growth is the main contributing factor. In addition to the gene transcripts, several classes of noncoding regulatory miRNA have been associated with this abnormality with a significant impact on fetal development and offspring health.

It is a well-established fact that increasing age incurred a pathophysiological condition to a growing human oocyte, which is mostly associated with aneuploidy [13,14]. This has led research on reproductive aging to focus on the importance of chromosomal abnormalities in reducing the developmental potential of female gametes. The biological reactions underlying aging are believed to occur spontaneously to give rise to cellular injuries with certain universality [15]. The most widely recognized biological reaction leading the modification of molecules during ovarian and oocyte aging is caused by oxidative stress [16]. The possible increase in the accumulation of reactive oxygen species (ROS) with aging in the ovarian environment can be attributed to reduced enzymatic antioxidant defense mechanisms [17]. Several studies have been conducted to investigate the mechanisms behind the age-associated oxidative stress damage and potential of antioxidant supplementation to avert those damages.

The non-coding miRNAs, which are active either in the cell or those released into the extracellular environment coupled with extracellular vesicles or proteins, have been evidenced to play a significant role in reproductive health. In this review data on the identification, characterization, and functional analysis of cellular and extracellular miRNAs in mammalian reproductive health associated with PCOS, endometritis, IUGR, and age-related disorders are presented.

## 2. MiRNAs and Polycystic Ovary Syndrome (PCOS)

Polycystic ovary syndrome (PCOS) is found to affect the ovarian follicular growth, granulosa cell proliferation and endocrine homeostasis [7,18]. Symptoms of PCOS include oligomenorrhea or amenorrhea, obstetrical complications, obesity, excess body and facial hair (hirsutism), acne, pelvic pain, low fertility, pregnant failure, and patches of thick and darker skin [18,19]. It is associated not only with infertility but also with increased risk of metabolic and other disorders including but not limited to insulin resistance, diabetes, obesity, mood disorders, obstructive sleep apnea, endometrial cancer, and cardiovascular diseases [7]. Complex genetic factors either heritable or non-heritable, which are evident at the onset of puberty, epigenetic, or/and environmental factors such as unhealthy diet and a lack of physical activity are known to contribute to the pathogenesis of PCOS [20]. Recent molecular genetic analyses including sequencing, expression profiling, and genome-wide association studies revealed the involvement of numerous miRNAs in the etiology of PCOS [21,22,23].

### 2.1. Expression and Regulation of Cellular miRNA in PCOS Ovary

Different animal models have been used to investigate the role of miRNAs in PCOS. A rat PCOS model has been established through the induction of dihydrotestosterone (DHT) to examine the expression of PCOS-associated ovarian miRNA [22]. The expression of miRNAs has also been examined in another study using the ovary of letrozole-induced PCOS rat through deep sequencing [21]. Compared to control rats, 129 and 44 miRNAs were found to be up- and downregulated in the ovary of PCOS rats, respectively. Wang et al. used a mouse model to study the role of miRNAs in the ovary with PCOS and increased expression of miR-27a-3p in the ovaries of mice with PCOS was observed [24]. Later on, using in vitro culture of primary mouse granulosa cells (mGCs) and the mouse granulosa-like tumor cell line it was found that miR-27a-3p represses CYP19A1 via targeting cyclic AMP response element (CRE)-binding protein 1 (Creb1) in granulosa cells from the PCOS mouse model [24]. The studies on miRNAs in the PCOS of the animal model mentioned above are mostly analysis of expression of miRNAs and no in depth functional study has been conducted yet. Moreover the results on the expression of miRNAs in the animal model with PCOS are not well matched to the findings (discussed below) from the studies in humans.

Expression of miRNAs in ovarian cortexes with PCOS was measured in a study using qRT-PCR and upregulation of miR-93 was observed [25]. The target gene of miR-93, CDKN1A was found to be downregulated in the ovary. Variation in the results of miRNAs expression profile among different studies might be due to the use of different models and methods, age of the animal, or women or inclusion criteria for the studies but could also exemplify the heterogeneous nature of PCOS itself.

Altered expression of ovarian miRNAs due to PCOS may have consequences on processes determining the fate of granulosa cells (proliferation, differentiation, and apoptosis). Ovarian granulosa and theca cells are the vital somatic cells that support follicular development; by producing and maintaining steroid, and sharing paracrine factors to favor oocyte growth and development. Extensive studies of miRNAs in granulosa cells both from normal ovaries [26,27,28] and ovaries with PCOS have been performed compared to that in other ovarian cell types. Expression and regulation of miRNAs in ovarian granulosa cell proliferation and apoptosis in various mammalian species are summarized in Table 1. A study on the expression of selected miRNAs in the granulosa cells of PCOS ovary by Cirillo et al. has identified the upregulation of MiR-146a, miR-155, miR-486, and downregulation of miR-320 and miR-370 in granulosa cells from PCOS patients [29]. Significantly increased expression of miR-200b was observed in the granulosa cells of PCOS patients compared to the controls revealed inhibition of cell proliferation through targeting PTEN [30].

**Table 1 ijms-22-00938-t001:** Expression and regulation of miRNAs in the granulosa cells with known target genes in relation to PCOS.

miRNAs	Species	Target Gene/Pathway	Biological Role	Expression in PCOS	Reference
miR-9	Human	IL8, SYT1, IRS2	Hinders testosterone release.	Up	[31]
miR-18b	Human	IL8, SYT1, IRS2	Promotes progesterone release, inhibitstestosterone and estradiol release	Up	[31]
miR-21	Human, Mouse, Rat	LATS1	Reduce apoptosis. Progression of PCOS	Up	[31,32,33,34]
miR-29a-5p	Human	Klotho-associated signaling	Involved in cell apoptosis	Down	[35]
miR-30c	Human		Induced by FSH exposure	Up	[34]
miR-93	Human	CDKN1A	Promotes cell proliferation	Up	[25]
miR-126-5p	Human	Klotho-associated signaling	Involved in cell apoptosis	Down	[35]
miR-129	Human	HMGB1	Proliferation, apoptosis of granulosa cells	Up	[36]
miR-132	Human Mouse Rat	HMGA2, Ctbp1	Promotes estradiol secretion, reduces progesterone and testosterone release	Down	[31,37]
miR-135a	Human	IL8, SYT1, IRS2	Reduces progesterone and testosterone release	Up	[31]
miR-145	Human	IRS1	Inhibits cell proliferation	Down	[38]
miR-155	Human	PDCD4	Prevents testosterone release, promotes cell proliferation and migration	Up	[34,39,40]
miR-222	Human Rat	Estrogen receptor 1	Positively correlated with serum insulin; increases estradiol secretion	Up	[22,41]
miR-224	Human Mouse	PTX3, Smad4	Induces GCs proliferation, increases estrogen release	Differentially expressed	[34,42,43]
miR-320	Human Mouse	RAB5B, E2F1, SF-1	Increased in insulin resistance, Slow down cell proliferation and estradiol production	Down in serum, Up in granulosa cells	[44,45,46]
miR-320a	Human	RUNX2	Related to the steroidogenesis	Down	[47]
miR-383	Human Mouse	RBMS1	Enhances the release of estradiol	Up	[46,48]
miR-483-5p	Human	Notch3, MAPK3, IGF1	Related to cell proliferation and apoptosis	Down	[23,49,50]
miR-509-3p	Human	MAP3K8	Induces oestradiol (E2) secretion	Up	[51]

Differential miRNA expression profiles of cumulus granulosa cells between the ovary of women with and without PCOS by next-generation sequencing revealed the upregulation of 21 and downregulation of 38 miRNAs [23]. In addition, through pathway analysis, it has been shown that Notch signaling pathway and the molecular pathways involved in regulation of hormones, and energy metabolism are targeted by the differentially expressed miRNAs. Both Notch3 and MAPK3, as members of Notch signaling and ERK-MAPK pathway, were demonstrated to be regulated by miR-483-5p. Downregulation of miR-483–5p and miR-486–5p was observed in the cumulus granulosa cells surrounding metaphase II oocytes of women with PCOS and the former was shown to be involved in the proliferation of granulosa cells of PCOS through induction of the PI3K/Akt pathway [49]. Similar to the granulosa cells, expression and regulation of miRNAs has been shown to be important in theca cells for the biosynthesis of androgen in ovaries. However, due to the difficulty to collect the theca cells from the PCOS ovary, studies on miRNAs in theca cells are limited compared to the granulosa cells. In situ hybridization of miRNAs in the ovary of a rat PCOS model revealed differential localization of miRNAs in the ovary undergoing or became cystic. Notably, miR-96, miR-31, and miR-222 were exclusively localized in the theca of cystic follicles [22]. Predominant localization of differentially expressed miRNAs in theca cells of the cystic ovary indicates their possible involvement in the abnormal androgenesis and hence might be important for the pathophysiology of PCOS. 

### 2.2. Extracellular miRNAs in the PCOS Ovary

Follicular fluid is the result of secretory and metabolic activity of the oocyte, granulosa, and theca cells and the transfer of blood plasma components. This specialized fluid provides an appropriate intrafollicular environment that plays a pivotal role in the growth and development of oocyte [52]. Changes of the content of follicular fluid have been used as predictive factors or biomarkers of oocytes and even embryo quality and PCOS development [53]. Follicular fluid is also used in determining the relationship between abnormal miRNA expression and PCOS development [45]. In addition, the presence of numerous miRNAs as extracellular nucleotide and changes of their expression in the follicular fluid from different species signifies the potential role of miRNAs in steroidogenesis and PCOS.

Quantification of miRNAs through TaqMan miRNA assays revealed a significantly lower expression of miR-132 in the follicular fluid of PCOS patients compared to healthy controls [45]. In another study, analysis of miRNAs of human follicular fluid revealed significant increased expression of five miRNAs (miR-32, miR-34c, miR-135a, miR-18b, and miR-9) in the PCOS group compared to the healthy control. Three potential target genes of these miRNAs were found to be significantly downregulated in the PCOS group and pathway analysis implicated their association with carbohydrate metabolism, beta-cell function, and steroid synthesis [42]. Significant downregulation of let-7b and miR-140 and upregulation of miR-30a was found in PCOS follicular fluid [53]. In the same study, the FOX-2 has been identified as a target of miR-30a and the disruption of miR-30a revealed excessive androgen production by theca cells and PCOS related morphological changes of granulosa cells. Further studies are needed to determine whether extracellular miRNAs play a role in the etiology of PCOS or if they are downstream effects of other PCOS risk factors [54]. 

The expression of miR-320 and miR-383 was upregulated in the follicular fluid of PCOS patients and the expression of E2F1 and SF-1 as a target gene of those miRNAs were found to be downregulated in granulosa cells [46]. These results suggest that miR-320 suppresses the granulosa cells proliferation and steroid production by targeting E2F1 and SF-1, which might be associated with hyperandrogenemia in PCOS. The miRNA profile in PCOS follicular fluid found to be varied among different studies due to differences in the methods employed, inclusion criteria used, type of PCOS diagnostic criteria, control groups set up, and heterogenic morphology and physiological conditions of PCOS [55].

Similar to the miRNAs in the follicular fluid, circulating serum miRNAs can also be promising and non-invasive biomarkers for different diseases. Changes in cellular homeostasis lead to the modulation of the molecular function of cells and deregulated miRNAs bound with protein component AGO2 or encapsulated with a microvesicle may be released from cells and frequently circulate in extracellular body fluids [56]. Extracellular vesicles released from most cell types into the extracellular space including serum can be taken up by the neighboring or distant cells to facilitate genetic exchange between cells [57,58]. Several studies have been conducted to identify miRNAs in serum from PCOS patients as a potential molecular signature. Investigation on the expression of circulating miRNAs in PCOS during clinical diagnosis revealed the upregulation of let-7i-3p, miR-5706, miR-4463, miR-3665, and miR-638 and downregulation of miR-124-3p, miR-128, miR-29a-3p, and let-7c in PCOS patients. Further analysis showed that the predicted target genes of these miRNAs are involved in various disorders, including diabetes and celiac diseases [59]. It has been demonstrated that the downregulation of miR-592 is associated with a high level of luteinizing hormone/chorionic gonadotropin receptor (LHCGR), an important factor for hyperandrogenemia in PCOS [60,61]. MiR-146a as a suppressor of steroid secretion was shown to be a negative regulator of serum testosterone in PCOS [44]. The same study identified that miR-222 is positively associated with serum insulin from PCOS patients. In addition, miR-222 combined with miR-146a and miR-30c suggested to be used to distinguish women with PCOS from the healthy ones [44]. Similar to the miR-146a, miR-23a in PCOS serum was reported to be negatively correlated with the testosterone level and the likelihood of PCOS [62]. 

## 3. Involvement of Cellular and Extracellular microRNAs in Endometritis 

During early postpartum stages, the uterine environment of dairy cows is susceptible to various uterine disorders [63], which reduces the reproductive and productive performance. Based on clinical signs, the nature and composition of uterine discharges and uterine cytology, uterine infections can be categorized as puerperal metritis, clinical metritis, clinical endometritis, and subclinical endometritis [64,65,66]. Endometritis is one of the leading uterine infections reported to impair the fertility of cows [67]. Endometritis is a phenomenon of endometrium inflammation without the manifestation of clear clinical signs. Multiple bacterial species including *E. coli*, *A. pyogenes*, *F. necrophorum,* and *B. melaninogenicus* are associated with endometritis, either individually or synergistically [68,69,70]. In bovine endometritis, the *E. coli* is one of the major pathogens that affect the uterine environment by producing a lipopolysaccharide (LPS) [71]. The initial defense mechanism against the pathogen invasion is through the activation of the animal’s innate immune system [72]. Following the activation of the innate immune system, the endometrial cells secrete cytokines and chemokines, which recruit neutrophils and macrophages to act on the pathogens [73]. In reaction to the pathogenic invasion, the neutrophils serve as the first line of defense and the number of the polynuclear neutrophils (PMN) is reported to increase in the uterine lumen [74]. As opposed to the clinical endometritis, the subclinical endometritis is characterized by inflammation of the uterus without visible clinical signs and an elevated number of PMN at a threshold of 5% and above [75].

Several studies have reported transcriptional changes resulting from uterine inflammation. The expression of inflammatory cytokines like the tumor necrosis factor-α (TNF-α), Toll-like receptor (TLR) 4, interleukins (IL), insulin-like growth factor-1 (IGF-1), and IGF-binding protein-2 (IGF-BP-2) were reported to be differentially expressed in the uterine of cows affected with endometritis [76,77]. In addition to the candidate genes screening approach, global transcriptome analysis in uterine cells obtained from cows with clinical and subclinical endometritis has been reported recently [78]. In that study about 203 and 28 genes were differentially expressed in cows with clinical and subclinical endometritis, respectively. This implies that the severity of infection significantly enhances the number of transcripts altered in the endometrial cells. Interestingly, these genes are involved in pathways related to the immune process, G-protein coupled receptor signaling, and chemotaxis.

MicroRNAs are thought to be involved in various uterine disorders [78,79,80]. Nevertheless, there are few in vivo experiments with respect to the expression profile of miRNAs in clinical and subclinical endometritis. Endometrial cytobrush samples derived from day 30 and 35 postpartum lactating cows with subclinical endometritis revealed 23 miRNAs to be differentially expressed in subclinical endometritis cows compared to the healthy cows [81]. In the same study, using an in vitro assay on epithelial and stromal cells, it was reported that the expression miR-24, miR-98, miR-223, miR-27, miR-196b, miR-503, miR-21, and miR-16 were upregulated in both epithelial and stromal cells treated with LPS. Moreover, the expression of miR-619, miR-215, and miR-643 was downregulated in response to LPS treatment. In another study, the miRNA expression of endometrial cytobrush derived from cows with clinical and subclinical endometritis at 40–60 days postpartum showed massive deregulation of miRNAs in both clinical and subclinical animals compared to the healthy counterparts [78]. As opposed to the less pronounced transcriptional deregulation impacts, the numbers of differentially regulated miRNAs were more pronounced in the subclinical groups than the clinical endometritis. This could be justified due to the negative transcriptional and posttranscriptional regulatory correlation between the miRNAs and the gene transcripts. Interestingly, 7 members of the let-7 family (let-7a, let-7c, let-7d, let-7d *, let-7e, let-7f, and let-7i) were among the highly abundant miRNAs preferentially enriched in the endometrial cells of the subclinical groups. Some miRNAs including let-7e, miR-92b, miR-337-3p, let-7f, and miR-145 were equally affected in both levels of the infection severity [78]. In addition to the genome-wide expression profiling of miRNAs associated with endometrial infection, understanding the functional implication of selected miRNAs during and post-infection could shed light to our understanding in the molecular mechanisms associated with etiology of the disease and development of diagnostic tools. 

Other important miRNAs involved in establishment and progression of endometritis is miR-148a [78], which is also implicated in inflammatory disease [82]. Bovine endometrial epithelial cells treated with LPS to induce an inflammatory response revealed a significant downregulation of both the miR-148a and the associated proinflammatory cytokines, IL-1ß and the TNF-α. Moreover, overexpression of miR-148a suppresses the activation of the NF-κBp56 pathway by targeted suppression of TLR-mediated pathway [83]. In the same study, it was shown that overexpression of miR-148a alleviated the uterine inflammation, making the miR-148a a potential therapeutic molecule for uterine inflammation and endometritis. MiR-223 is another miRNA reported to be abundantly enriched in cows with subclinical endometritis and in endometrial cells treated with LPS [81]. Consistent with this finding, using an in vitro cell culture model of bovine endometrial epithelial cells treated with LPS, the transcription of miR-223 was elevated, which was promoted through the activation of the NF-κB and inhibition of the NLRP3, which in turn mediated the production of the cytokine IL-1ß [84]. This signifies the role of miR-223 in attenuating inflammatory conditions in the uterine environment. MiR-19a, which is a member of the miR-19-72 cluster, regulates the expression of TBK1 by negatively regulating the NF-κB pathway in LPS-induced endometritis [85]. These findings suggest that the proinflammatory pathways are negatively regulated by miRNAs expressed in endometrial epithelial cells.

The biggest challenge in the diagnosis and treatment of uterine infections is the absence of non-invasive, reliable, and early diagnostic tools that can be used for early detection of the disease. Thus, the use of circulating miRNAs in animals could be an alternative approach. To address this, a pilot experiment on the clinical endometritis cow model was performed to determine the unique circulatory miRNAs associated with uterine infection. Briefly, lactating dairy cows were subjected to vaginal infusion of *E. coli* and *T. pyogenes*. The induction of uterine infection was verified according to previously described criteria [86] and the blood samples were collected and processed at days 2, 6, 9, 14, and 21 post-infection. The genome-wide miRNAs expression profile was assessed and compared with the preinfection status of the cows. Interestingly, no measurable differences were observed in the circulatory miRNA profile of cows until day 14 post-infection. However, starting day 14, the miRNA expression was more divergent, showing the sustained and severe endometrial infection (Figure 1). This was further verified by the hierarchical clustering analysis, where the number of upregulated miRNAs tends to be progressively increased starting day 14 towards day 21 post-infection (Figure 2). Taken together, the progression of endometrial infection can be diagnosed by measuring circulatory miRNAs as early as two weeks post infection. In a separate experiment, the serum circulatory miRNA profile in cows with underlying metritis and with no apparent uterine abnormality derived 1 week postpartum was performed. It was reported that bta-miR-15b, bta-miR-17-3p, bta-miR-16b, bta-miR-148a, bta-miR-26b, bta-miR-101, and bta-miR-29b were upregulated whereas bta-miR-148b, bta-miR-199a-3p, bta-miR-122, bta-miR-200b, and bta-miR-10a were downregulated in the serum of cows with metritis compared to healthy cows [87].

Extracellular vesicles (EVs) are involved in immune activities affecting both the innate and adaptive immune responses [88], which are also mediated by miRNAs encapsulated in the EVs [89,90]. Recently, the correlation of between exosome-coupled miRNAs derived from serum of patients with endometritis and healthy counterparts revealed 24 miRNAs differentially expressed, of which the miR-320a and miR-22-3p, were significantly upregulated in exosomes of patients with endometritis and can be potential biomarkers of endometritis [91]. EV-derived miRNAs from the uterine fluid of cows had a negative impact on the blastocyst development when added to the embryo culture media [92]. Contrary to this, supplementation of embryos with EVs derived from conditioned embryo culture media and uterine fluid of healthy cows improved the cleavage rate and blastocyst formation [93,94]. It was also shown that miR-218 was one of the differentially expressed miRNAs and involved in the pathogenesis of bovine endometritis and bovine endometrial cells are reported to use the EVs to encapsulate and release the miR-218 into the uterine environment, which acts as an inhibitor of immune factors like the IL-6, IL-1b, and TNFα and inflammatory genes [95]. This signifies the fact that EV-coupled miRNAs could be one of the mechanisms for delivering immune regulating molecules in the uterus. 

In conclusion, the current advancements in the EVs research and high-throughput sequencing technologies have provided the promising tools for the discovery and screening of clinical and subclinical endometritis with better precision and as early as possible, which could improve the fertility and welfare of animals. Moreover, the possibility of enrichment, loading, and delivering a bioactive molecule of our interest including miRNAs and proteins into cells could be a possible approach to understand the mechanistic molecular roles of selected miRNAs or proteins in endometrial health.

## 4. MicroRNAs in Intrauterine Growth Restriction

Intrauterine growth restriction (IUGR) is a pregnancy-related disorder in which a fetus cannot reach its maximum growth potential and the weight of the fetus is less than the 10^th^ percentile of an appropriate weight for gestational age and fetal sex. Although different maternal and fetal factors can contribute to the development of IUGR, impaired placental development and function is the most common cause [96]. In recent years, miRNAs have emerged as major players in the pathogenesis of IUGR. miRNAs can be easily measured in tissue biopsies, blood, and other biological samples, making them potential biomarkers for early detection of various pathologies including IUGR [97,98,99]. Trophoblast proliferation and invasion is critical for placental health, therefore most of the miRNAs linked to trophoblast proliferation are also involved in the pathogenesis of IUGR [100].

Some unique species of miRNAs are exclusively expressed in human trophoblast cells and are called “trophomiRs” [101]. The majority of trophomiRs are expressed from chromosome 19 miRNA cluster (C19MC) [102]. C19MC locus generates 59 mature miRNAs, most of which are primarily expressed in the placenta [103,104]. Hromadnikova et al. demonstrated a correlation between C19MC miRNAs and pathogenesis of pregnancy complications by measuring the expression of 15 C19MC miRNAs in placental tissue from pregnancies complicated by preeclampsia (PE), IUGR, and gestational hypertension (GH) [105]. They reported that downregulation of six C19MC miRNAs (miR-517-5p, miR-518f-5p, miR-519a, miR-519d, miR-520a-5p, and miR-525) is associated with IUGR. Additionally, reduced expression of miR-517-5p, miR-519d, miR-520a-5p, and miR-525 is a common finding in PE, IUGR, and GH, whereas downregulation of miR-518f-5p and miR-519a is a common finding in preeclampsia and IUGR [105]. In contrast, according to Hromadnikova et al. increased circulating levels of C19MC miRNAs (miR-517-5p, miR-518b, and miR-520h) during the first trimester indicate increased risk of PE, but no circulating C19MC miRNAs can be used as biomarkers to predict IUGR [106]. Chromosome 14 miRNA cluster (C14MC) produces 63 mature miRNAs in humans, which are highly expressed in the placenta [107]. C14MC miRNAs play a critical role in placental development by regulating the expression of different genes involved in trophoblast proliferation, invasion, and migration [108,109]. From the first trimester to the third trimester of pregnancy in humans, the concentration of C19MC miRNAs increases whereas the concentration of C14MC miRNAs decreases in the placenta and maternal circulation [110,111,112]. Wommack et al. measured different miRNA clusters in maternal plasma and showed that increased expression of several C14MC miRNAs (miR-3373p, miR-4315p, miR-1363p, miR-1365p, miR-3803p, miR-323a3p, and miR-543) can be used as a marker for birth weight and gestation length [112].

Circulating placenta-specific miRNAs including miR-520a, miR-520h, miR-525, miR-526a, miR-516-5p, miR-517, and miR-518b in maternal plasma from IUGR pregnancies are significantly increased at week 12-16 of gestation, which later comes back to normal levels [110]. The temporary increase in these miRNAs can cause trophoblast apoptosis resulting in placental insufficiency and IUGR. Increased maternal plasma concentration of miR-206 can be used to predict IUGR [113]. miR-206 targets vascular endothelial growth factor (VEGF), which is critical for placental development. In a study conducted using 820 pregnant women (74 IUGR and 746 non-IUGR pregnancies), Li and Liu measured miR-206 and VEGF concentrations in maternal plasma at different stages of pregnancies. miR-206 was significantly increased whereas VEGF was significantly reduced in maternal blood at early, middle, and late gestation [113]. These findings suggest that miR-206 can lead to IUGR by targeting VEGF and its higher levels in maternal blood can be used to predict IUGR at the early stages of pregnancy.

miR-141 targets proliferation-associated genes and is downregulated in different types of cancers [114]. Tang et al. showed that miR-141 is associated with IUGR and its expression in placental tissue from IUGR pregnancies is higher compared to control pregnancies [115]. Moreover, expression of miR-141 target genes including E2F transcription factor 3 (E2F3) and pleiomorphic adenoma gene 1 (PLAG1) is significantly reduced in IUGR placentas [115]. E2F3 and PLAG1 are involved in cell proliferation and reduced expression of these genes can cause impaired trophoblast proliferation and contribute to the pathogenesis of IUGR [115]. According to Mouillet et al. the concentration of trophoblastic-hypoxia induced miRNAs (miR-27a-1, miR-30d, miR-93, miR-141, miR-200c, miR-205, miR-224, miR-335, mir-424, miR-451, and miR-491) is increased in maternal circulation with IUGR pregnancies compared to normal pregnancies [116]. Huang et al. also reported the role of miR-424 in the pathogenesis of IUGR by demonstrating that miR-424 is significantly increased and its target genes, mitogen-activated protein kinase 1 (MEK1) and fibroblast growth factor receptor 1 (FGFR1), are significantly reduced in placentas from IUGR pregnancies [117].

Using small RNA next-generation sequencing in fetal and maternal placental tissues, Rutkowska et al. demonstrated that 48 fetal and 22 maternal miRNAs are differentially expressed in IUGR pregnancies compared to normal pregnancies [118]. The change in miRNAs with a predicted role in fetal growth and development (miR-29a, miR-92b, miR-125b, miR-7641, miR-4321, miR-let-7g, and miR-2895) was also confirmed by RT-PCR [118]. The maximum change was witnessed in miR-4321, which was 4-fold higher in fetal placenta from IUGR pregnancies compared to control pregnancies [118]. An important miR-4321 target gene involved in placental and fetal growth is prostaglandin E receptor 3 (PTGER3), which has high expression in human trophoblast cells and plays a role in trophoblast invasion [118,119]. However, the concentration of miR-4321 in maternal circulation has never been studied, making it hard to use it as a potential biomarker for noninvasive diagnosis of IUGR.

The let-7 family of miRNAs (let-7a, let-7b, let-7c, let-7d, let-7e, let-7f, let-7g, let-7i, and miR-98), also referred to as differentiation-inducing miRNAs, play a profound role in trophoblast proliferation [120]. The expression of let-7 miRNAs is suppressed by oncoprotein LIN28, which has two paralogs LIN28A and LINB [121,122]. In 2019, Ali et al. demonstrated that the term human placentas from IUGR pregnancies have reduced expression of LIN28A and LIN28B and higher expression of let-7 miRNAs (let-7a, let-7b, let-7c, let-7d, let-7e, let-7f, let-7g, and let-7i), indicating a correlation between IUGR and let-7 miRNAs [120]. In trophoblast cells, let-7 miRNAs target several genes involved in cell proliferation, cell invasion, and angiogenesis including AT-hook 1 (*HMGA1*), MYC protooncogene (*c-MYC*), vascular endothelial growth factor A (*VEGF-A*), Wnt family member 1 (*WNT1*), AT-rich interaction domain (ARID)3A, and ARID3B [100,120,123]. CRISPR-Cas9 based LIN28 knockout in human trophoblast cells leads to increased expression of let-7 miRNAs and reduces cell proliferation [120]. Trophectoderm-specific knockdown of LIN28A or LIN28B in sheep increases let-7 miRNAs and reduces trophoblast cell proliferation and conceptus elongation in vivo [123]. Reduced conceptus elongation in domestic ruminants has been linked to IUGR and early pregnancy loss. Collectively, these findings indicate that a higher level of let-7 miRNAs in trophoblast cells can contribute to the pathogenesis of IUGR primarily by causing impaired placental development.

Awamleh et al. measured miRNAs and gene expression in human chorionic villi from pregnancies complicated by PE, IUGR, or PE+IUGR using next-generation sequencing. They showed that 11 miRNAs were upregulated in PE placenta samples, 25 miRNAs were upregulated, 12 miRNAs were downregulated in IUGR placentas, and 9 miRNAs were upregulated PE + IUGR placentas [124]. Similarly, 275 genes were differentially expressed in PE placentas, 155 genes were differentially expressed in IUGR placentas, and 556 genes were differentially expressed in PE + IUGR placentas. Six differentially expressed miRNAs (miR-193b-5p, miR-193b-3p, miR-210-3p, miR-365a-3p, miR-365b-3p, and miR-520a-3p) were common in all groups [124]. Hromadnikova et al. examined the expression of 32 miRNAs in placentas from pregnancies complicated with PR and IUGR that needed to be terminated before week 34 of gestation. They showed that 11 miRNAs (miR-16-5p, miR-100-5p, miR-122-5p, miR-125b-5p, miR-126-3p, miR-143-3p, miR-195-5p, miR-199a-5p, miR-221-3p, miR-342-3p, and miR-574-3p) were downregulated in pregnancies complicated by IUGR without PE [125]. Together these findings provide a set of miRNAs that are dysregulated in pregnancies complicated by just IUGR and can be used as potential biomarkers for the diagnosis of IUGR.

## 5. Potential of miRNAs in Aging and Related Disorders

The reproductive organ of the mammalian female exhibits a rate of aging that is much faster than any other organs in the body system. It has been speculated that the premature aging of the ovary, when compared with the somatic organs, might result from increased demand of energy for the maintenance and repair processes in the soma compartment during aging [126]. Based on the human biologic clock, the loss in female fertility becomes dramatic after the age of 35 years and results in menopause at 50–51 years of age [127]. The ovarian functional decline due to aging is related to the gradual loss of resting follicles and decreased biological competence of those surviving age-related atresia [128]. Throughout the life of female, follicles leave the resting pool of primordial stage to the growing pool on a regular basis and pass through various stages under the influence of stage-specific intraovarian regulators and endocrine factors [129,130]. This oocyte and follicle pool decline with increasing age, with a marked increase in the rate of disappearance from the age of 37 to 38 onwards. At the stage of menopause the follicle reserve decline to a number insufficient to sustain the cyclic hormonal process necessary for menstruation [131]. Moreover, in addition to the reduction in the follicular reserve, the phenomenon of ooplasm aging has been reported to be associated with various structural and morphological abnormalities including chromosome decondensation, chromosomal misalignment associated with anomalies in meiotic spindle, and highly compromised cellular machinery [132,133]. The proportion of poor quality oocytes increases with age, ranging from 50% at 20 years of age to 95% at 35 years of age [134]. All in all, decreased female fertility with advanced maternal age is well documented and it is widely recognized that the decline in oocyte quality is a key factor to explain age-associated infertility problems and related high risk of birth defects, genetic disorders and miscarriage [135,136]. Gene expression studies in oocytes showed that the presence and activity of gene products involved in cell cycle regulation, spindle formation, and organelle integrity may be altered in oocytes from older individuals from several species. Transcriptome analysis of human MII oocytes in relation to aging and ovarian reserve demonstrated that there is differential expression of coding and noncoding transcripts between young and old women [137]. Moreover, analysis of global gene expression profiles of MII oocytes from young (<35 years) and older (>37 years) women identified differential expression of genes associated with cell cycle regulation, cytoskeletal and chromosomal structure, energy pathways, transcription control, and stress response [138,139]. The differential expression of protein coding mRNA was found to be accompanied by the expression of noncoding regulatory RNA, including lncRNA, piRNA, and precursor miRNAs in oocytes between young and old women [137]. Similarly, transcriptome analysis of GV stage oocytes from young and aged mice showed differential expression of 160 endogenous small-interfering RNAs and 10 miRNAs [140]. 

Appropriate storage and utilization of maternal transcripts in oocytes are needed for maturation and early embryo development. A study by [141] showed that human MII stage oocytes obtained from women of advanced reproductive age showed altered expression of miRNAs regulating gene expression, pluripotency, chromatin remodeling, and early embryo development. In the same study evolutionary conserved miRNA (miR-29a-3p, miR-203a-3p, and miR-494-3p), which were found to be upregulated in aged mouse oocytes, to be correlated with downregulation of DNMT3a, DNMT3b, phosphatase and tensin homolog (PTEN), and mitochondrial transcription factor A (TFAM).

Cellular senescence is the biological consequence of aging, implicated in a variety of age-associated diseases. The increased vulnerability of oocytes to age-induced oxidative stress is associated with the attenuation of the efficacy of DNA double-strand break (DSB) repair mechanisms in aged oocytes [142]. Some studies have been conducted to correlate follicular fluid extracellular miRNA with advanced women age. MicroRNA expression profile of follicular fluid from younger (<31 years) and older (>38 years) women revealed a set of miRNAs involved in heparan-sulfate biosynthesis, extracellular matrix–receptor interaction, carbohydrate digestion, and absorption, p53 signaling, and cytokine–cytokine–receptor interaction [143]. With increased sample size and reduced age gap, the study by [144] revealed a differential expression of a single miRNA (hsa-mir-424). Differential plasma expression of miRNAs have been identified in cattle in an age- and genetic-dependent manner [145]. Using the PCR array platform, 306 plasma miRNA were assessed between calf and mature cows, and 26 miRNAs including miR-192, miR205, and miR-215, were enriched in mature animals. 

Cellular senescence, which imposes permanent proliferative arrest in cells in response to stressors, is considered as a hallmark of aging and a major risk factor for the development of most common age-related diseases (ARDs) and it is an attractive target for therapeutic applications [146]. Senescence cells are characterized by a significantly reduced replicative potential and by the acquisition of a proinflammatory senescence-associated secretory phenotype (SASP) [147]. Considerable efforts have been devoted to distinguishing the effects of several epigenetic mechanisms namely: DNA methylation, and long and small non-coding RNA on the transcriptional and posttranscriptional programming leading to cellular senescence. Senescence modulation by microRNAs is a major senescence-associated epigenetic mechanism. This has been suggested by cellular miRNA signatures and by the release of extracellular vesicles, which contain different species and a number of miRNAs and proteins. The EVs population and type seem to reflect the molecular characteristics of their cells of origin and modulate the phenotype of recipient cells [148]. A recent study conducted to unravel the relative contribution of EVs released from senescence cells in spreading prosenescence signals to proliferating cells via their miRNA cargo [149]. In the same study it has been shown that senescence human umbilical endothelial cells release a greater number of EVs compared to their control counterparts. Among the 22 miRNAs differentially expressed, miR-21-5p and miR-217 were found to be enriched both in cells and EVs of senescence cells and are found to target DNMT1 and SIRT1 genes. Coculture of EVs from senescence with control cells resulted in induction of the senescence phenotype as evidenced by modulation of DNMT1 and SIRT1 genes, apoptosis, and cellular proliferation. 

Considering mares as the best model to study reproductive aging in humans, a study on the identification of extracellular vesicles coupled miRNAs in relation to aging would provide knowledge on the mechanisms behind age-related fertility problems [150]. To investigate the effect of mare age on exosomal miRNA expression during follicular development, follicular fluid exosomes isolated from the normal follicle at deviation, mid-oestrus, and preovulatory stage of young (3–12 years) and old (20–26 years) were subjected to miRNA analysis [151]. In that study, exosomal miRNA expression differences were observed both across the developmental stages and between age groups. The abundance of miR-513a-3p, miR-181A, and miR-375 was higher in exosomes derived from follicular fluid of old compared to young mares. Among these, miR-181A was found to negatively regulate mouse granulosa cell proliferation by targeting ACVR2A [152].

Postovulatory oocyte aging in vitro or in vivo demonstrated that those aged oocytes frequently showed lower fertilization rate, polyspermy, chromosomal abnormalities, and abnormal embryo development [153]. These abnormalities in early embryo development result in decreased litter size, lower pregnancy rates, and an increase in the number of spontaneous miscarriages in humans [154]. In vitro and in vivo postovulatory aging oocytes are known to exhibit various cellular and molecular changes associated with intracellular signaling. These include morphological and organelle changes, reduction in the intracytoplasmic level of antioxidant GSH, elevated reactive oxygen species (ROS), reduction in the intracytoplasmic level of adenosine triphosphate (ATP), decrease in the expression of antiapoptotic factor Bcl-2, increased apoptosis, and abnormal Ca^2+^ regulation (reviewed by [155]). As a consequence of the progressive increase in ROS accumulation and the concomitant decline in antioxidant protection, the postovulatory aged oocytes experienced the state of oxidative stress. It has been long thought that the oxidative stress may act as a trigger for a cascade of factors that orchestrate postovulatory aging. Several in vitro studies have demonstrated that supplementation of antioxidants attenuates the process of postovulatory aging, however, there is a great variation between various antioxidants applied. This may be associated with the specificity of various antioxidants in scavenging the different types of reactive oxygen species. Considering the fact that cells exposed to various environmental stressors are known to release vesicles enriched with antioxidants [148], there is a great potential of supplementing such vesicles during in vitro oocyte maturation, which could lead to prevention of postovulatory oocyte aging phenotypes.

## 6. Conclusions

Reproductive disorders are a major cause of fetal and maternal morbidity and mortality in humans and cause huge economic losses to the cattle and sheep industry. Early diagnosis of these disorders can help in better management and treatment. Although several environmental and physiological factors can contribute to the pathophysiology of reproductive disorders, microRNAs have emerged as major players in the reproductive health of animals. One miRNA can target hundreds of different genes and hence several molecular pathways involved in reproductive health and efficiency of mammals are also regulated by miRNAs. In this review we summarized the miRNAs, which are differentially expressed in various reproductive disorders, suggesting the role of these miRNAs in pathogenesis of different reproductive disorders. Although the exact cause of miRNA dysregulation is unclear, epigenetic modifications, random genetic mutations, adverse uterine environment, oxidative stress, and malnutrition are some of the possible factors, which can cause dysregulation of miRNAs. MiRNAs are secreted by different cells and tissues in the extracellular environment either directly or via vesicles. MiRNAs can be detected and readily measured in different biological samples including peripheral blood, tissue biopsies, saliva, cerebrospinal fluid, and urine. Due to this reason, dysregulation of specific miRNAs can be used as a biomarker for early diagnosis of different reproductive disorders. As described in this review, dysregulation of more than one miRNA have been linked to various reproductive disorders. Therefore, using miRNAs as a biomarker for early diagnosis of different health conditions can be challenging and further studies are needed to identify the miRNAs that can be used as reliable biomarkers for each health condition.

## Figures and Tables

**Figure 1 ijms-22-00938-f001:**
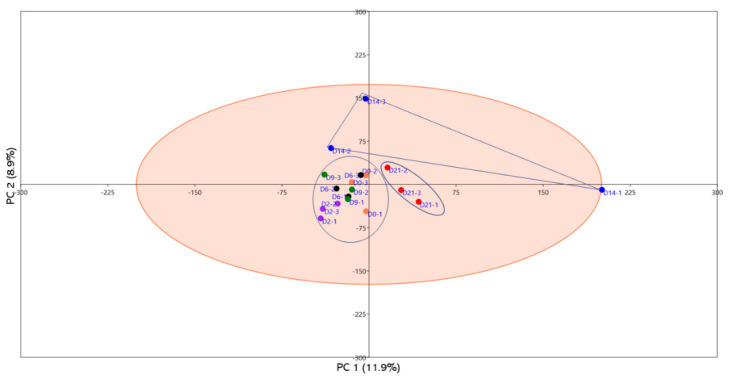
Principal component analysis (PCA) of circulatory miRNAs of blood plasma samples derived from cows subjected to bacterial infection. Samples were collected at day 0 (orange dotes) pre-infection and days 2 (purple dotes), 6 (black dotes), 9 (green dotes), 14 (dark blue dotes), and 21 (red dotes) post-infection. It is shown that the groups from days 14 (dark blue dotes) and 21 (red dotes) showed clear separation from the earlier post-infection periods. Groups are indicated by different color in biological triplicates.

**Figure 2 ijms-22-00938-f002:**
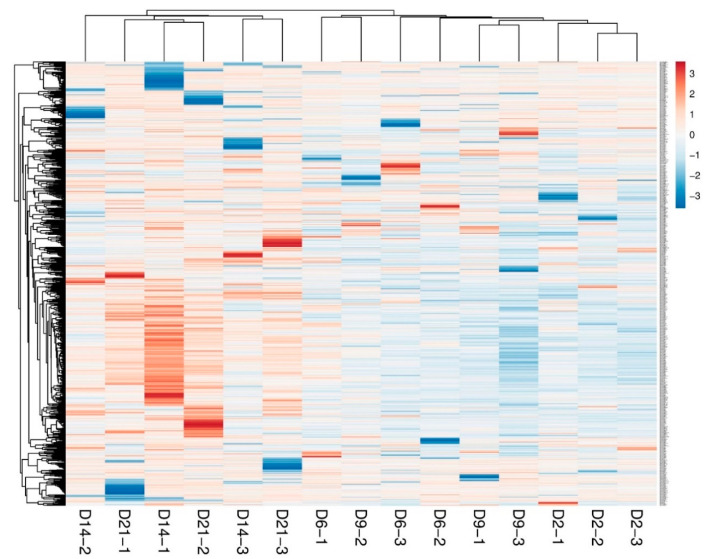
Heatmap illustrating the hierarchical clustering of samples according to the expression pattern of the plasma circulatory miRNAs profile. Groups are indicated in biological triplicates.

## Data Availability

The data that support the findings of this study are available from the corresponding author upon reasonable request.

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
