# Peer review of "MicroRNA-Mediated Gene Regulatory Mechanisms in Mammalian Female Reproductive Health"

_ijms, 2021, doi:10.3390/ijms22020938_

Round 1

Reviewer 1 Report

Overall, the authors revised the manuscript according to the reviewers' suggestions.  One suggestion (very minor) is that instead of justify the biological roles column in Table 1, it would be easier to read if the authors align the text with the left margin.

Author Response

Overall, the authors revised the manuscript according to the reviewers' suggestions.  One suggestion (very minor) is that instead of justify the biological roles column in Table 1, it would be easier to read if the authors align the text with the left margin.

Response: The column in the table is left-justified as recommended by the reviewer.

Reviewer 2 Report

Please, see the attached file

  What do you want to do ? New mailCopy   What do you want to do ? New mailCopy

Author Response

 I recommend its publication with only minor modifications, mainly in the table 1 and figure 1.

  1. Table 1: it should be presented on a single page for better readability.

Response: Because of the long list of miRNAs and their associated information, it cannot be accommodated in one single page. This might be possible during the final production process of the paper.

  1. In table 1: the separation between columns 4 (Biological Role) and 5 (Expression PCOS) should be greater and the alignment of the text in row 4 must be changed to make the data easier to read.

Response: The Biological role column is now left justified and the column 5 is centered so that the information from column 4 and 5 clearly separated.

  1. In table 1, row 9: with respect to the reference 36, the mir-129 seems to be regulated upwards in POCs, not downwards.

Response: Thanks for correction. We have changed it to up regulated.

  1. In table 1, row 15: the meaning of DE needs to be explained.

Response: The term DE is fully written as differentially expressed.

  1. In figure 1: the percentage of variability of the two PCA axes should be added.

Response: The percentage of variability is added in the figure.

  1. In figure 1: the legend of the colored dots shall be added and the two separate groups shall be highlighted with an elliptical circle.

Response: The color dotes designated each days of sample collection included in the PCA analysis are described in the figure legend. Those clustered together are highlighted with circle or triangle lines.

Reviewer 3 Report

The review is very interesting and very good structurate. I suggest few minor language and editing corrections.

Point 1. The abstract needs to be modified because its not easy to follow and avoided abbreviators. 

Point 2. Specify in the introduction part how autophagyand miRNA is regulation in many cancers. what are results of these investigations, how are they correlated with malignancy?

Author Response

The review is very interesting and very good structurate. I suggest few minor language and editing corrections.

Point 1. The abstract needs to be modified because its not easy to follow and avoided abbreviators. 

Response: All abbreviations in the abstract are described when they appear for the first time.

Point 2. Specify in the introduction part of how autophagy and miRNA is regulation in many cancers. what are the results of these investigations, how are they correlated with malignancy?

Response: Despite the fact that the reviewer raised an important aspect of the role of miRNAs in cancer biology, describing the mechanism in which miRNAs plays a role in autophagy and malignancy will be out of the scope of the present review.